# Mean Dietary Salt Intake in Vanuatu: A Population Survey of 755 Participants on Efate Island

**DOI:** 10.3390/nu11040916

**Published:** 2019-04-24

**Authors:** Katherine Paterson, Nerida Hinge, Emalie Sparks, Kathy Trieu, Joseph Alvin Santos, Len Tarivonda, Wendy Snowdon, Jacqui Webster, Claire Johnson

**Affiliations:** 1World Health Organization, Vanuatu Office, Ministry of Health Iatika Complex, Cornwall Street, Port Vila, Vanuatu; 2Vanuatu Ministry of Health, Iatika Complex, Cornwall St, Port Vila, Vanuatu; itahinge@gmail.com (N.H.); ltarivonda@vanuatu.gov.vu (L.T.); 3The George Institute for Global Health, The University of New South Wales, Sydney 2006, NSW, Australia; esparks@georgeinstitute.org.au (E.S.); ktrieu@georgeinstitute.org.au (K.T.); jsantos@georgeinstitute.org.au (J.A.S.); jwebster@georgeinstitute.org.au (J.W.); cjohnson@georgeinstitute.org.au (C.J.); 4World Health Organization, Division of Pacific Technical Support, South Pacific Office, Level 4, Provident Plaza One, Downtown Boulevard, 33 Ellery Street, Suva, Fiji; snowdonw@who.int

**Keywords:** humans, sodium chloride, salt, Vanuatu, non-communicable diseases, prevention and control, nutrition policy

## Abstract

Non-communicable diseases are responsible for 63% of global deaths, with a higher burden in low- and middle-income countries. Hypertension is the leading cause of cardiovascular-disease-related deaths worldwide, and approximately 1.7 million deaths are directly attributable to excess salt intake annually. There has been little research conducted on the level of salt consumption amongst the population of Vanuatu. Based on data from other Pacific Island countries and knowledge of changing regional diets, it was predicted that salt intake would exceed the World Health Organization’s (WHO) recommended maximum of 5 g per day. The current study aimed to provide Vanuatu with a preliminary baseline assessment of population salt intake on Efate Island. A cross-sectional survey collected demographic, clinical, and urine data from participants aged 18 to 69 years in rural and urban communities on Efate Island in October 2016 and February 2017. Mean salt intake was determined to be 7.2 (SD 2.3) g/day from spot urine samples, and 5.9 (SD 3.6) g/day from 24-h urine samples, both of which exceed the WHO recommended maximum. Based on the spot urine samples, males had significantly higher salt intake than females (7.8 g compared to 6.5 g; *p* < 0.001) and almost 85% of the population consumed more than the WHO recommended maximum daily amount. A coordinated government strategy is recommended to reduce salt consumption, including fiscal policies, engagement with the food industry, and education and awareness-raising to promote behavior change.

## 1. Introduction

High salt intake increases blood pressure, which is recognized as a risk factor for non-communicable diseases (NCDs), in particular cardiovascular diseases (CVDs), including heart attack and stroke [1]. Sixty-three percent of global deaths are attributable to NCDs, with a higher burden in low- and middle-income countries [2]. In 2015, there was an estimated 17.9 million deaths related to CVDs worldwide, with high blood pressure being the leading cause [3]. Further, CVD-related deaths attributable to excess salt intake have been estimated at 1.7 million deaths annually worldwide [4].

A reduction in population salt intake is one of the single most effective public health strategies to reduce the burden of NCDs worldwide [5], prompting the World Health Assembly to adopt a population salt intake reduction target of 30% by 2025, in 2013 [2]. Salt reduction has also been named as one of the World Health Organization’s (WHO) NCD “best buy” strategies, which defines interventions that are affordable, feasible, and cost-effective [6], as reducing salt intake lowers blood pressure, and thereby the risk of CVDs [1]. The WHO recommends a reduction in salt intake to less than 5 g per day to prevent NCDs [7]. Globally, populations are consuming excessive amounts of salt, with the worldwide estimated mean salt intake being almost 10 g/day, double the recommendation [8].

Vanuatu is a lower middle-income country in the Pacific Islands [9] and has a high prevalence of deaths from NCDs, estimated at 70% in 2008, with 36% attributable to CVDs [10]. The 2013 WHO STEPwise approach to surveillance of non-communicable disease risk factor (STEPS) survey reported almost 30% of the Vanuatu population had raised blood pressure or was on medication for high blood pressure [11]. High blood pressure and CVDs place a large economic burden on the government: it was estimated that the average inpatient hospital expenditure on admissions related to arterial diseases was approximately $2516 USD per person per stay in 2012 [12]. It has also been estimated that the government could save up to US $75 per person per year in pharmaceutical costs if hypertensive people were able to control their blood pressure through lifestyle choices [13].

The rise in NCDs in many of the Pacific Island Countries (PICs) occurred during the latter half of the twentieth century, coinciding with the changing dietary patterns of the region [14]. There has been a shift away from the traditional diets of the PICs, consisting primarily of root crops, starchy fruit and vegetables, seafood, and coconut products, toward a more Western-style diet, including refined cereals and grains, edible fats and oils, processed foods and higher quantities of land animals [14]. Today, high salt containing foods are commonly consumed in PICs, including bread, instant noodles, packaged snack foods, such as crisps and crackers, canned foods, and salty sauces, such as soy sauce [15]. Furthermore, as the availability of processed foods and commercially prepared meals increase, salt intake is also likely to increase [16,17]. The use of discretionary salt added during cooking and at the table may also be increasing as this practice becomes more common [15].

The increased accessibility and availability of high salt foods is suggested to be a contributor to the high prevalence of NCDs found in Vanuatu [4]. While increased salt intake has previously been a topic of discussion [11,18], it has never before been a government priority, and as such, there is not yet a population measurement of salt intake for Vanuatu. Therefore, the main objective of the current study was to provide Vanuatu with a preliminary baseline assessment of population salt intake on Efate Island, estimated using spot urine samples and validated with measured 24-h salt excretion from a subsample. Whilst 24-h urine samples are the gold standard methodology [19], spot urine samples validated by a subsample of 24-h urine collections are often utilized in low resource settings, such as Vanuatu [20], in order to obtain a viable result. The secondary aim was to identify the proportion of the population with salt intake above the WHO recommended maximum intake of 5 g per day. Based on data from Samoa and Fiji, which show population salt intake exceeding the WHO’s recommend maximum of 5 g per day [21,22], and knowledge of changing diets in the Pacific [14], it was predicted that salt intake in Vanuatu would also exceed the WHO’s recommended salt consumption limit.

## 2. Materials and Methods

This cross-sectional survey collected data from rural and urban communities on Efate Island during two time periods, October 2016 and February 2017, due to administrative delays part way through the survey. Ethical approval was obtained through the Vanuatu Ministry of Health. Written informed consent was obtained from all participants as well as the freedom to withdraw from the survey at any time.

### 2.1. Survey Population and Sampling

The population was divided into two strata (urban and rural) by mapping of Enumeration Areas (EA), which was previously undertaken by the Vanuatu National Statistics Office (VNSO) [23]. A total of 28 EAs were randomly selected using the WHO STEPS sampling frame [24], including 14 urban and 14 rural areas.

During the October 2016 data collection period, 50 households were listed in the selected EAs and 27 were randomly selected using the lottery method [25]. Participant selection was undertaken via a convenience sampling method, such that eligible household members at home at the time of interviewing were invited to participate in the survey. Participants were eligible to participate in the study if they were aged between 18–69 years and not pregnant or menstruating at the time of urine collection. Only one person from each household was interviewed. During the February 2017 data collection, the number of households selected in each EA increased from 27 to 36 due to a low response rate in 2016. In 2017, both households and participants were selected via convenience sampling due to lower than expected participant and household response rates. Low response rates have been previously documented as a challenge for data collection in the Pacific [21].

### 2.2. Data Collection

The survey was conducted in communities on Efate Island, the most populous island in Vanuatu. As migration from outer islands to Efate is high, the population of Efate can be seen as somewhat representative of the country’s population. Local field researchers were appropriately trained in order to undertake data collection, as per the WHO STEPS survey protocol [24]. These research methods have also been used successfully in neighboring PICs, including Samoa and Fiji [21,22]. All questionnaires, consent forms, information sheets, and urine collection instructions were translated in Bislama, the national language of Vanuatu.

Demographic data was collected via a questionnaire and included questions relating to age, education level, gender, employment, NCD risk factors, and history of disease. Clinical data, including recording of weight, height, blood pressure, and measurement of sodium and potassium levels by spot and 24-h urine samples, were also obtained.

Weight was measured in kilograms to the nearest 100 g using digital bathroom scales on a flat surface and height was measured in centimeters to the nearest millimeter using a stadiometer [24]. Blood pressure was measured using the OMRON Automatic Blood Pressure Machine, in millimeters of mercury (mmHg). Three measurements were performed on the participant’s left arm with 3-min intervals in between. Final blood pressure was calculated as an average of the second and third measurements. High blood pressure was defined as systolic blood pressure greater than 140 mmHg or diastolic blood pressure greater than 90 mmHg.

Spot urine samples were collected from each participant after they completed the interview. All participants were given a 100 mL spot urine container and estimated mean salt intake was calculated using the INTERSALT with potassium equation, validated in the estimation of mean population levels of 24-h sodium and potassium excretion [26,27,28].

A random subsample of participants were asked to provide a 24-h urine sample. Participants were given verbal and written instructions for the process in Bislama. They were asked to discard the first void of urine collected in the morning and provide the time that this occurred, and then begin collection with the following urine onwards for the next 24 h, including the first urine of the following morning. Participants were asked to provide the time of the final urine and report any issues with collection, such as missed collection or spillage. Urine collected was stored in a provided 5-L container and participants were asked to store it in a cool, dry area, with the lid on tight. The urine samples were collected by field researchers one to two days after completion.

Participants who did not meet the eligibility criteria, had incomplete demographic data, or suspected inaccurate urine collection based on the urine volume and urinary creatinine excretion, were excluded [22]. The process of determining the final participant number is outlined in Figure 1.

### 2.3. Biochemical Assessment

The spot and 24-h urine samples were analyzed to determine the level of sodium, creatinine, and potassium at the Laboratory Department at Vila Central Hospital in Port Vila using a Cobas C311 urine analyzer.

### 2.4. Statistical Analyses

Statistical analyses were undertaken by The George Institute for Global Health in Sydney, Australia. All analyses were weighted to reflect the age, sex, and area distribution of the Efate population based on the 2016 mini census [23]. Statistical testing was conducted in STATA 13 for Windows (StataCorp LP, Texas), with a significance level of *p* < 0.05.

## 3. Results

A total of 774 participants completed the survey, of which 541 provided spot samples and 121 provided 24-h urine samples. After applying the criteria in Figure 1, 483 and 71 participants had complete spot and 24-h urine samples, respectively, and were included in these analyses, giving a final participation rate of 62.4%.

The mean age was approximately 37 years and 49% of all participants were female. Most participants were urban dwellers (66%) and just over half (53%) were employed. The majority of participants reported completing either primary (44%) or secondary schooling (42%; Table 1). Sixty percent of the participants were found to have a body mass index (BMI) in the overweight or obese category (66% in women versus 55% in men). The mean BMI was 27.3 kg/m^2^ (overweight), and men had a lower mean BMI than women (26.6 compared to 28.1 kg/m^2^). Mean blood pressure was 122/78 mmHg. Only 10% of the participants reported having previously been diagnosed with hypertension, however 21.6% were found to have raised blood pressure, with a further 2.4% taking anti-hypertensive medication (Table 1).

### Estimated Salt Intake from Urine Samples

Mean salt intake was estimated to be 7.2 (SD 2.3) g/day derived from spot urine samples. Males had significantly higher salt intake than females (Difference 1.3 g/day; 95% CI 0.9 to 1.7 g/day; *p* < 0.001). Almost 85% of the population consumed more than the WHO’s recommended amount of 5 g salt per day, with 87.6% of men and 81.1% of women exceeding the guideline. Mean salt intake using 24-h urine samples was found to be 5.9 (SD 3.6) g/day, and no sex differences were determined (Table 2). There was no statistical difference observed between the characteristics of participants who completed a spot urine and 24-h urine sample (Appendix A). A Bland-Altman plot showed all but two data points (97%) fell within the limits of agreement (−5.8 and 9.1; +/− 2SDs of the mean difference of 1.6 g/day), indicating that there is little variability between the results of the spot and 24-h urine for those participants who provided both samples (*n* = 65; Appendix B).

## 4. Discussion

The Vanuatu Salt Intake Survey is the first cross-sectional study to provide a baseline estimate of salt intake using urinary analysis. The majority of the study population, almost 85%, consumed more than the WHO recommended amount of 5 g, and 8% of the population consumed over 10 g/day; double the guideline. Estimated mean salt intake was found to be 7.2 g/day from spot urine samples in the sub-national study population, with men consuming significantly more salt than women. Mean salt intake using 24-h urine samples was found to be 5.9 g/day, 1.3 g/day less than the spot urine estimate.

The characteristics of participants who provided spot urine samples were not significantly different to those providing 24-h urine samples, suggesting it is possible to estimate mean salt intake from spot urines. The sample size of 24-h urine samples was small, and therefore estimates may not be precise, as indicated by the large standard deviation. However, the smaller standard deviation given from spot urine samples is likely a result of the equation used, which incorporates age, sex, and BMI; variables which are relatively constant, and thereby generate less variability [29]. As there was a larger sample of spot urine samples, this may be a more precise estimate, however this method is known to overestimate population mean intake at lower levels and underestimate at higher intake levels [29,30], and has not yet been validated in this population. The WHO recommends the collection of spot urine samples in countries where low capacity and resources impede the ability to collect 24-h urine samples, after validation of spot urine in the population [20]. Our intention was to validate the spot urine samples with the 24-h urine samples as per the recommended procedure, however, the low 24-h urine sample size prevented this. Taking all of these factors into account, we can surmise that the result from the spot samples in this population may reflect a more accurate measurement of population salt intake under these circumstances. Future research will still require a sub-sample of 24-h urine collection to validate spot urine samples and may provide a better estimate.

While an exact estimate of mean salt intake cannot be obtained from spot urine samples in this study, it is known that spot urine estimates are able to accurately classify population salt intake as above or below the 5 g/day maximum salt target set by the WHO [30]. From this, we can conclude that the Vanuatu population are consuming excess amounts of salt and salt reduction strategies should be a priority for the country.

Mean salt intake from 24-h urine samples was similar to previous estimates in Vanuatu. The 2010 Global Burden of Diseases (GBD) modelling study estimated salt intake as 5.6 g/day in Vanuatu [31], and the Household Income and Expenditure Survey (HIES), 2010, estimated mean salt intake as 5.2 g per person per day [32]. However there are notable shortcomings of HIES data, which include underestimation of staple foods and overestimation of infrequently consumed foods [33] and processed foods, which tend to be high in salt [34]. While the 24-h urine estimate from this study is similar to these previous estimates, the spot urine samples suggest salt intake may be higher than previously recorded.

Salt intake estimated from spot urine was 7.2 g/day, which was similar to a nationally representative sample in Samoa in 2013 (7.09 g/day), though this was measured from 24-h urine samples [35]. The similarity of these results is highly likely given the demographic, environmental, and cultural similarities between the two countries [9,36]. Likewise, gender differences were also found in this study (male salt intake was significantly greater than female), mirroring the Samoa study [35]. In contrast, salt consumption was lower than Fiji’s most recently recorded salt intake data from 24-h urine samples in 2016 [21]. This may be due to the increased urbanization and differences in food culture in Fiji, for example Fijian Indian cuisine having a high salt content [37].

Financial constraints resulted in multiple study limitations. Due to limited financial resources and administrative delays, a common occurrence in the Pacific, the sampling was conducted at two time points, which may have induced seasonal bias. Further, the method of sampling was also modified from random sampling of households in 2016 to convenience sampling in 2017 due to a lack of finances, personnel, and time restraints. However, the estimated intakes from both samples are likely comparable [38]. Financial limitations meant that the study was only conducted in one island of Vanuatu, Efate Island, which was chosen because it has the highest population density and rate of urbanization of all the islands.

The greatest strength of this study was the ability to determine the proportion of the population with a salt intake of greater than 5 g/day. This research was also the first to provide an estimate of population salt intake from urinary analysis and provides a baseline from which to monitor population salt intake. Furthermore, the protocol used is detailed enough to be replicated and improved upon for future data collection. An additional strength is the inclusion and empowerment of local people. The data collection teams were entirely comprised of, and led by, ni-Vanuatu, and all questionnaires and procedures were culturally sensitive and inclusive towards ni-Vanuatu people and their customs. This fostered a sense of community ownership and encouraged participation. Given the limited health research previously carried out in Vanuatu, and lack of local people’s exposure to research of this kind, it is vital that the importance of this research is conveyed to people in order for the results to be implemented. As such, we ensured that outcomes of this research were appropriately disseminated to the relevant ministries and also the general population through local media, creating conversation and awareness about salt intake. This research has highlighted the need for action to reduce salt consumption in Vanuatu, where previously there was little awareness in the general population of a need for change.

Future research in the region should anticipate the difficulties in data and urine sample collection in the Pacific, including low response rates, and plan for a larger number of survey teams collecting data from more communities. Additionally, raising awareness of research within communities should be extensive to increase and support participation. It should be comprised of varied media forms, including TV, radio, and newspapers, in addition to speaking to, and involving, chiefs and community leaders.

### Recommendations

A large proportion of the Vanuatu population are consuming more than 5 g of salt per day. This baseline assessment of salt intake in Vanuatu provides the required evidence for the implementation of a national salt reduction initiative. Previous interventions have demonstrated that a multi-sectorial, multi-factored approach is most effective in lowering population salt intake [39]. Components of this approach might include fiscal policies, such as a salt tax, engagement with food industry to reduce salt levels in foods, and education and awareness-raising interventions to promote behavior change regarding low salt food choices and discretionary salt use [40]. To appropriately direct these components, additional research will be needed, particularly the identification of major sources of salt in the diet, which can be obtained by conducting two 24-h diet recalls [41,42]. Results from a knowledge, attitudes, and behavior questionnaire conducted at the same time-points as the urine collection could also be used to identify population subgroups that should be targeted for education and awareness-raising campaigns [42]. Furthermore, the SHAKE the Salt Habit Technical Package for Salt Reduction [43] provides instruction on the most effective ways to develop, implement, and monitor salt reduction strategies and could be a useful tool for Vanuatu to achieve a reduction in population salt intake via a variety of methods.

Due to the multi-factorial nature of NCDs, prevention and control strategies need the partnership and engagement of multi-sectoral stakeholders both inside and outside the health sector, including relevant government ministries, non-government organizations, the food industry, private sector, and the communities themselves. Future actions point to political commitment on the part of the government to mobilize resources for salt reduction programs, while additionally creating a monitoring plan to track changes in salt intake, which could be completed through the already established WHO STEPs protocol [24].

## 5. Conclusions

This is the first study to assess salt intake in Vanuatu using urine samples and confirms that the majority of the Vanuatu population are consuming salt intakes greater than the WHO recommended maximum intake. Introducing a national salt reduction strategy could decrease salt intake, or prevent the likely increase in salt consumption, and reduce the burden of NCDs in Vanuatu. Future studies in Vanuatu and other PICs should consider the challenging environment and other lessons learned from this study to improve research efficiency and effectiveness.

## Figures and Tables

**Figure 1 nutrients-11-00916-f001:**
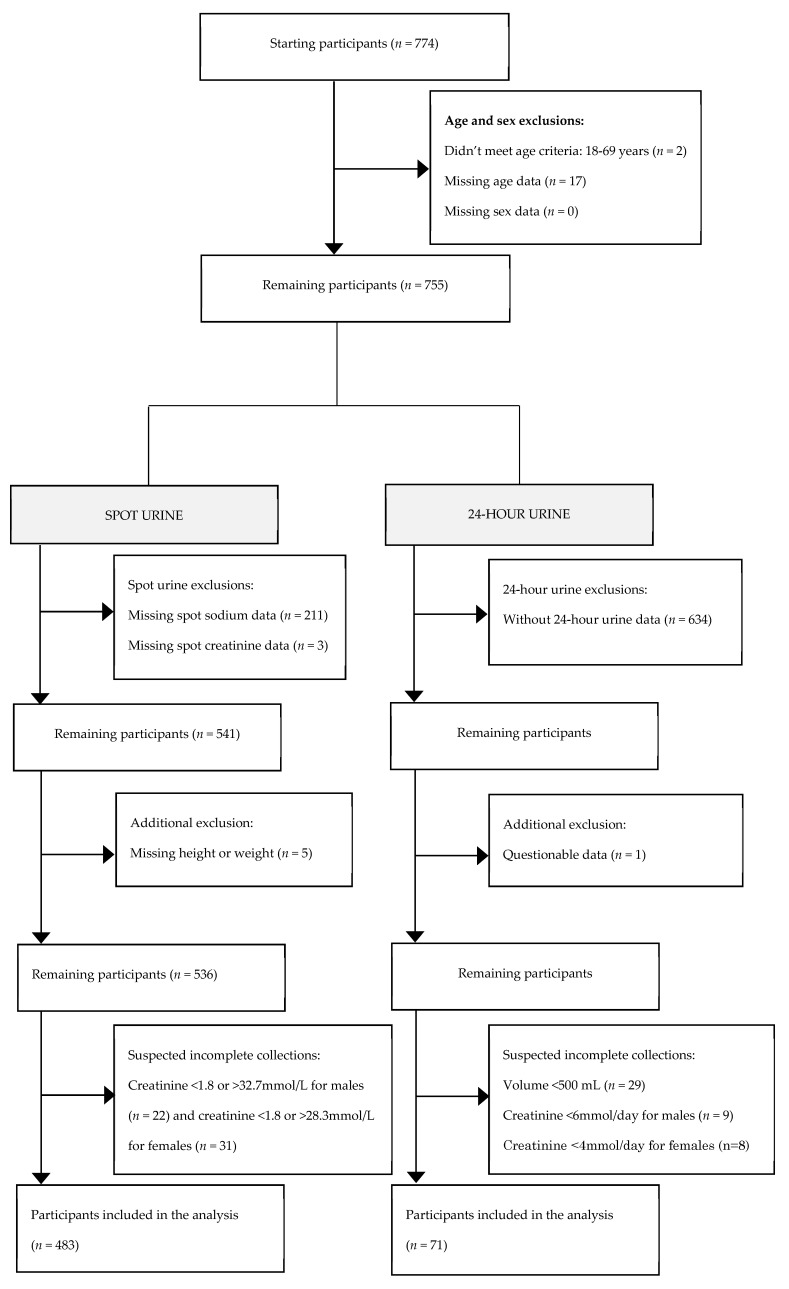
Process of determining the final participant number.

**Table 1 nutrients-11-00916-t001:** Weighted sample characteristics.

Characteristics	Overall	Female	Male
Age in years, mean (SD)	36.6 (12.6)	35.6 (12.3)	37.5 (12.8)
Female, %	49.2	-	-
Area, %			
Rural	34.1	34.7	33.4
Urban	66.0	65.3	66.6
Completed education, %			
No formal schooling	2.5	2.8	2.2
Primary level	43.6	47.9	39.6
Secondary level	41.7	38.7	44.7
Tertiary level	12.1	10.7	13.5
Employed, %	52.8	43.7	61.6
Height in cm, mean (SD)	164.1 (7.9)	158.9 (5.7)	169.1 (6.3)
Weight in kg, mean (SD)	73.6 (15.6)	71.1 (16.3)	76.0 (14.6)
Body mass index in kg/m^2^, mean (SD)	27.3 (5.4)	28.1 (5.9)	26.6 (4.8)
Overweight or obese, %	60.3	54.8	65.9
Systolic blood pressure in mmHg, mean (SD)	121.7 (19.2)	116.5 (18.1)	126.8 (19.0)
Diastolic blood pressure in mmHg, mean (SD)	77.9 (12.7)	76.4 (11.4)	79.3 (13.8)
History of hypertension, %	10.3	10.9	9.8
Measured hypertension, %	21.6	16.1	26.9
Pre-existing hypertension (those who had measured high blood pressure OR were taking Western hypertension medication prior to the survey), %	24.0	18.8	29.0
History of high cholesterol in blood, %	3.7	5.1	2.4
History of heart attack, %	1.8	0.8	2.9
History of stroke, %	1.1	0.4	1.7
History of diabetes, %	2.0	1.0	3.1
History of chronic kidney disease, %	1.8	1.0	2.6

**Table 2 nutrients-11-00916-t002:** Weighted results for salt intake.

	*n*	Overall	Female	Male	*p*-Value
24-h urine in g/day, mean (SD)	71	5.9 (3.6)	5.6 (3.7)	6.2 (3.5)	0.496
Spot urine using “INTERSALT with potassium” equation in g/day, mean (SD)	483	7.2 (2.3)	6.5 (1.7)	7.8 (2.6)	<0.001
Salt intake above the 5 g WHO target, %	483	84.4	81.1	87.6	

WHO, World Health Organization.

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
