# Peer review of "Mean Dietary Salt Intake in Vanuatu: A Population Survey of 755 Participants on Efate Island"

_nutrients, 2019, doi:10.3390/nu11040916_

Reviewer 1 Report

Limitations:

1- the sample population average age is around 35-36 years old which means the elderly adults who have higher prevalence of hypertension did not include in this study in the proportion that reflects general population.

2- it would be better to include a diet history (24 hour recall, food frequency list) to understand the role of the dietary components in sodium intake.

3- The sample size for 24 hour urine is too small

Minor correction: 

L 183: BMI stands for body mass index not body weight index.  

Author Response

Dear Reviewer,

We would like to thank you for your comments on the above named manuscript. Please find a response to all comments in the attached Word document.

Kind regards,

Katherine Paterson

Reviewer 2 Report

This is a manuscript reporting results from a study examining mean dietary salt intake on the Vanuatu population. The manuscript is concisely presented, which is a strength, but it also lacks sufficient justification for its significance.

I recommend you show the differences and/or correlations between estimated and measured 24-h urinary sodium excretion among the participants examined sodium excretion from both spot and 24-hr urine (I think the number of participants is 71). And, what do you think about creating new formulas in predicting 24-hr urinary sodium excretion from spot urine in Vanuatu population as below?

A review article (ref. #20): “Therefore it is recommended that in those providing spot urine that a sub-sample also perform 24 h urine collections to enable the development of valid estimating equations that can provide a robust mean population estimate of sodium intake from spot collections”

How about using Kawasaki and Tanaka formulas additionally in predicting 24-h sodium excretion from spot urine?

Author Response

Dear Reviewer,

We would like to thank you for your comments on the above named manuscript. Please find attached a response to all comments in the attached Word document.

Kind regards,

Katherine Paterson

REVIEWER 2

Reviewer comments

Author’s response

Changes made to manuscript

1.This is a manuscript reporting results from a   study examining mean dietary salt intake on the Vanuatu population. The   manuscript is concisely presented, which is a strength, but it also lacks   sufficient justification for its significance.

Thank you for your comment.   We were unclear on whether you meant the significance of the study, the   results or the future directions, so we have addressed all three below.

Alterations have been made   to the introduction to emphasize the importance of this research.

The significance of the   results are presented in the discussion (lines 208-211, 260-263, 281-283),   and significance for future interventions are presented in lines 283-284 and   301–304.

Lines 82 – 85: changed from   “Based on data from other PICs [21, 22]   and knowledge of changing regional diets [14],   it was predicted that salt intake would exceed the World Health   Organization’s recommended maximum of 5g per day.” to “Based on data from Samoa and Fiji which show population salt intake   exceeding the WHO’s recommend maximum of 5g per day [21, 22] and knowledge of   changing diets in the Pacific [14], it was predicted   that salt intake in Vanuatu would also exceed the WHO’s recommended salt   consumption limit.”

2.I recommend you show the differences and/or correlations between   estimated and measured 24-h urinary sodium excretion among the participants   examined sodium excretion from both spot and 24-hr urine (I think the number   of participants is 71). And, what do you think about creating new formulas in   predicting 24-hr urinary sodium excretion from spot urine in Vanuatu   population as below?

Thank you for your   suggestions. We have now included a Bland-Altman plot in Appendix B to show   the differences between the 24-hour and spot urine results for participants   who had both samples taken. We have also inserted an explanation into the   results section 3.1 (lines 201-204).

We feel that creating a new   formula for predicting 24-hour urinary sodium excretion from spot urine in   the Vanuatu population wouldn’t be accurate using our dataset for several   reasons. Firstly, we believe the 24-hour urine sample was too small (noted by   reviewer one), with the small size preventing us from validating the spot urine   sample (lines 217-229). Secondly, we made the assumption that the Efate   population was somewhat representative of the Vanuatu population (lines   110-111), however we would need to conduct further research to confirm this   before we considered making a new equation for the population.

Lines 201-204: insertion of A Bland-Altman plot revealed all but two data   points (97%) fell within the limits of agreement (-5.8 and 9.1; +/- 2SD of   the mean difference of 1.6g/day) indicating that there is little variability   between the results of the spot and 24-hour urine for those participants who   provided both samples (n=65; Appendix B).”

Lines 331-335: Insertion of Appendix B -   Bland-Altman plot and caption

“Figure 2. Bland-Altman   plot for 24-hour sodium excretion estimated from spot urine using the   INTERSALT with K method”

3.A review article (ref. #20): “Therefore it is   recommended that in those providing spot urine that a sub-sample also perform   24 h urine collections to enable the development of valid estimating   equations that can provide a robust mean population estimate of sodium intake   from spot collections”

Thank you for your comment.   As above, our intention was to collect enough 24-hour urine samples to be   able to validate the spot urine samples, however, due to limitations   discussed within the manuscript (lines 217-229) our sample size was too small   to accurately do this. In addition, a more recent review and meta-analysis of   spot and 24-hour urine data (Huang et al. 2016, ref 30) found that spot urine   estimates are able to accurately classify population salt intake as above or   below the 5 g/day maximum salt target set by the WHO, which we have   presented.

Nil

4.How about using Kawasaki and Tanaka formulas   additionally in predicting 24-h sodium excretion from spot urine?

We treated the INTERSALT with potassium equation as the   main spot urine result as it is the most commonly applied spot urine formula   (Huang 2016 – ref 30), however as suggested we tested other equations and   these are presented below. All spot urine equations showed salt intake   exceeded the recommended maximum of 5g/day. The Mage equation produced the   closest estimate to the 24-hour urine sample, however the large SD indicates   it may not be the best formula to use in this population. The INTERSALT with   K equation produced the next closest estimate, with a smaller SD (Table 2).

Nil

TABLE. Weighted results from other spot equations

Equation

Salt intake,   g/day (mean, SD)

N

Overall

Female

Male

Tanaka equation

483

7.6 (2.2)

7.6 (2.2)

7.5 (2.2)

Kasawaki   equation

483

9.5 (3.6)

8.9 (3.1)

10.1 (3.9)

Mage

483

6.4 (6.5)

6.3 (5.4)

6.5 (7.4)

Toft equation

483

8.6 (2.3)

7.3 (1.0)

9.8 (2.5)

INTERSALT   without K equation

483

7.4 (2.2)

6.7 (1.6)

8.0 (2.5)

Round  2

Reviewer 2 Report

Congratulations!!!

I accept your manuscript in present form.